# Active Learning with Low-Rank Structure for Data Selection

**Vincent Cohen-Addad** [1]  **Sasidhar Kunapuli** [2]  **Vahab Mirrokni** [1]  **Mahdi Nikdan** [3]
**David P. Woodruff** [1 4]  **Samson Zhou** [5]

## Abstract

In the data selection problem, the objective is to choose a small, representative subset of data that can be used to efficiently train a machine learning model. Sener and Savarese [ICLR 2018] showed that, given an embedding representation of the data and suitable geometric assumptions, heuristics based on $k$-center clustering can be used to perform data selection. This perspective was further explored by Axiotis et. al. [ICML 2024], who proposed a data selection approach based on $k$-means clustering and sensitivity sampling. However, these methods rely on the assumption that the dataset exhibits intrinsic geometric structure that can be effectively captured by clustering, whereas many modern datasets instead possess global algebraic structure that is better exploited by low-rank approximation or principal component analysis.

In this paper, we introduce a new data selection framework based on low-rank approximation and residual-based sampling, formulated through the lens of row subset selection and loss-preserving coreset construction. Given an embedding representation of the data satisfying mild regularity conditions, which can be interpreted as algebraic or angular notions of Lipschitz continuity, we show that it is possible to select a weighted subset of $\tilde{O}\left(k + \frac{1}{\varepsilon^2}\right)$ data points whose average loss approximates the average loss over the full dataset within a $(1 + \varepsilon)$ relative error, up to an additive $\varepsilon\Phi_k$ term, where $\Phi_k$ denotes the opti-

mal rank-$k$ approximation cost of the embedding matrix. We complement these theoretical guarantees with empirical evaluations, demonstrating that on a range of real-world datasets, our data selection approach achieves improved performance over prior strategies based on uniform sampling or clustering-based sensitivity sampling.

## 1. Introduction

The unprecedented growth of both datasets and models has fueled the success of modern machine learning, culminating in foundation models with remarkable capabilities across domains. Yet, this success comes at a steep cost: training and fine-tuning these models requires immense computational resources, extensive datasets, and long training cycles, rendering the process nearly impossible for most academic groups and small-scale companies. Importantly, however, it is now well understood that using the entire dataset is rarely necessary, as carefully chosen subsets of data often suffice to achieve nearly the same performance, with only a marginal increase in error. This observation raises a fundamental and urgent question:

> How can we efficiently identify the most informative subset of training data without compromising model quality?

While uniform sampling can often perform reasonably well in practice, it is inherently suboptimal on complex, imbalanced, or redundant datasets (Chawla et al., 2002; He & Garcia, 2009). To better capture the utility of individual data points for training, a large body of work on *data selection and active learning* aims to identify examples that are most informative given their uniqueness, quality, or relationship to the model's current knowledge. Although no universally optimal active learning strategy exists (Dasgupta, 2004), many heuristics have proven effective in practice (Lewis & Catlett, 1994; Tong & Koller, 2001b; Settles, 2009; Ren et al., 2020). Active learning is typically framed as an iterative process: a model is trained, then used to score and select a subset of unlabeled points for annotation. State-of-the-art methods rely on *uncertainty-based criteria*, such as margin or entropy, which prioritize points on which the model is

*Equal contribution  [1]Google Research  [2]University of California, Berkeley  [3]Institute of Science and Technology Austria (ISTA)  [4]Carnegie Mellon University  [5]Texas A&M University. Correspondence to: Vincent Cohen-Addad <cohenaddad@google.com>, Sasidhar Kunapuli <sasidhar.kunapuli@gmail.com>, Vahab Mirrokni <mirrokni@google.com>, Mahdi Nikdan <nikdanmahdi@gmail.com>, David P. Woodruff <dwoodruf@andrew.cmu.edu>, Samson Zhou <samsonzhou@gmail.com>.

*Proceedings of the 43$^{rd}$ International Conference on Machine Learning*, Seoul, South Korea. PMLR 306, 2026. Copyright 2026 by the author(s).

least confident (Brinker, 2003). However, when applied at scale, such strategies face two key limitations. First, computing selection scores requires evaluating the model on the entire dataset, which is computationally prohibitive for modern large-scale architectures. Second, practical training pipelines, especially those involving CNNs or foundation models, require acquiring and processing data in *batches* rather than one point at a time. This induces correlations among selected samples, which substantially reduces the effectiveness of uncertainty-based heuristics and limits their impact on training efficiency.

A major step forward was made by (Sener & Savarese, 2018), who reframed active learning as a coreset selection problem. Informally, a *coreset* is a small, carefully chosen subset of the dataset that approximately preserves the key properties of the full data (Har-Peled & Mazumdar, 2004; Chen, 2009; Feldman & Langberg, 2011; Langberg & Schulman, 2010; Phillips, 2016; Mirzasoleiman et al., 2020; Feldman, 2020; Braverman et al., 2021a), i.e., if a model or algorithm performs well on the coreset, it should perform nearly as well as if it had seen the entire dataset. Coresets are particularly valuable in modern machine learning because they reduce both memory and computational requirements while maintaining strong approximation guarantees. The insight of Sener & Savarese (2018) was that the difficulty of applying uncertainty-based methods in modern training pipelines stems from two central obstacles. First, training must proceed in *batches*, not one example at a time. Effective batch acquisition requires both informativeness and diversity, yet diversity often conflicts with standard objectives such as margin maximization, leading to redundant or near-duplicate selections (Citovsky et al., 2021). Second, computing uncertainty scores requires running inference over the entire dataset, which is already prohibitive for CNNs and becomes intractable for current large-scale architectures (Lowell et al., 2019).

To overcome these barriers, Sener & Savarese (2018) proposed to directly seek a small subset of data that serves as a coreset: training on the coreset should yield nearly the same model as training on the full dataset. In formal terms, the gradients (or losses) averaged over the coreset should approximate those of the entire dataset, so that optimization on the subset faithfully reproduces the effect of optimization on all data. Since computing these gradients exactly is impractical, they introduced a geometric relaxation: given an embedding representation of the data, one can approximate the coreset by solving a variant of the classical $k$-center problem. Subsequently, Axiotis et al. (2024) explored other coreset constructions based on other $k$-clustering objectives. These formulations are both natural and widely applicable, as embeddings can be obtained from pretrained encoders such as BERT (Devlin et al., 2019a), word2vec (Mikolov et al., 2013), GloVe (Pennington et al., 2014), ResNet (He

et al., 2016), or CLIP (Radford et al., 2021). Empirically, these approaches delivered state-of-the-art results in image classification benchmarks, demonstrating that geometric coreset selection can substantially outperform uncertainty-based heuristics in batch training scenarios.

Unfortunately, while embedding-based selection methods based on $k$-center and its $k$-clustering variants can be effective in some settings, they often exhibit critical limitations in modern machine learning applications. In high-dimensional datasets, clustering focuses on local groupings of points and can fail to capture the dominant directions of variance, meaning that selected subsets may miss the most informative components of the data. Low-rank approximation methods, by contrast, explicitly aim to preserve these dominant directions, ensuring that small subsets retain the essential spectral structure of the dataset. This phenomenon has been recently observed in the context of Low-Rank Adaptation (LoRA) (Hu et al., 2022; Xu et al., 2024; Wu et al., 2024; Li et al., 2024), where low-rank updates capture the most important components of the parameter space and enable efficient adaptation, whereas naive clustering of embeddings may overlook key directions. These observations suggest that, while clustering retains value in some cases, low-rank-based selection can provide a more reliable foundation for data-efficient training on large-scale models. Indeed, for many foundational tasks in data analysis and machine learning, the central loss function can be expressed as a *low-rank approximation* objective. Examples include principal component analysis (PCA), matrix completion, and dimensionality reduction.

## 2. Methodology and Contributions

Fine-tuning a Large Language Model (LLM) for a specialized task, such as translation, can be extremely costly when using the entire dataset, even if ample data is available. In practice, it is often preferable to select a small subset of points that preserves the essential structure of the data while still allowing the model to achieve high performance. Directly computing data importance, for example through model-based loss or margin scores, is typically impractical because it requires evaluating every data point with the full LLM, which is computationally expensive.

In this work, we propose a framework for *data selection under low-rank losses* that addresses this challenge by combining accurate but costly scores on a small fraction of the data with fast-to-compute embeddings or sketches that capture the dominant directions of the dataset. Surprisingly, simple embeddings derived from pre-trained models, such as BERT (Devlin et al., 2019a) or word2vec (Mikolov et al., 2013), are often sufficient to approximate the low-rank structure relevant for selecting informative points, even for much larger target models. We construct a low-rank sketch of

the dataset to estimate leverage scores, which informally quantify the importance of each point with respect to the orthogonal space of the sketch, and then sample rows proportionally to these scores. This ensures that the selected subset reflects the main directions of variance, rather than merely promoting diversity as in clustering-based coresets.

A key insight of our work is that effective data selection can be achieved by prioritizing the low-rank structure of the dataset, rather than relying solely on geometric diversity as in previous clustering-based approaches, which often selected points to cover the data space uniformly or to maximize distances between samples (Sener & Savarese, 2018; Axiotis et al., 2024). These approaches implicitly assume that distance between points is more important than the directions of the points. In contrast, our framework focuses on the dominant spectral directions captured by a low-rank sketch, ensuring that the selected subset represents the principal axes of variation and preserves the essential structure of the loss function. This perspective allows us to combine efficiency and theoretical guarantees with improved empirical performance, particularly in high-dimensional settings where clustering alone may overlook directions that are critical for model training.

By focusing on preserving the dominant spectral components, this framework offers a robust and efficient alternative to both naive subsampling and clustering-based data selection in high-dimensional and large-scale settings. Beyond the theoretical guarantees, the approach is simple, scalable, and broadly applicable. We demonstrate its effectiveness empirically on both a standard tabular dataset and challenging Llama3-8B (Dubey et al., 2024) fine-tuning on three tasks, outperforming the accuracy of existing baselines.

Before describing the theoretical guarantees for our approach, we first describe a number of natural assumptions for reasonable loss functions on datasets that are well-captured by algebraic properties. Let $V = \text{span}\{v_1, \ldots, v_k\}$ be a $k$-dimensional subspace, for instance corresponding to the top singular vectors, principal components, or basis directions from a low-rank factorization. For any point $y \in \mathbb{R}^d$, decompose

$$y = \alpha_1 v_1 + \ldots + \alpha_k v_k + r(y), \quad r(y) = \text{Proj}_{V^\perp}(y),$$

where $\alpha_i = \langle y, v_i \rangle$ and $r(y)$ is the component orthogonal to $V$, i.e., the projection of $y$ onto $V^\perp$. Let $v(y) = \text{Proj}_V(y) = \alpha_1 v_1 + \cdots + \alpha_k v_k$.

**Assumption 2.1.** *We assume there exist constants $\lambda, \gamma > 0$ such that*

$$|\ell(y) - \ell(v(y))| \leq \lambda \|r(y)\|_2^2,$$

$$\left| |\ell(v(y)) - \sum_{j=1}^{k} \alpha_j^2 \ell(v_j)| \right| \leq \gamma \sum_{i=1}^{k} |\alpha_i^2 + 1| \, \ell(v_i).$$

Informally, this condition decomposes the loss at $y$ into two components: a weighted sum of the losses along each basis direction $v_i$, with weights $\alpha_i^2$, and a penalty proportional to the squared norm of the component orthogonal to $V$, $\|r(y)\|_2^2$. Generally, when the scaling $\lambda$ for the orthogonal components is significantly larger than the scaling $\gamma$ for the subspace components, Assumption 2.1 intuitively suggests that the loss function has worse approximations in directions outside of the subspace, as we do not have information about these factors.

**Interpretation of the assumption.** Assumption 2.1 should be interpreted as a regularity and approximation condition rather than an exact structural requirement on the loss function. The first inequality asserts that deviations of a point $y$ outside the subspace $V$ incur loss that grows at most quadratically in the distance to $V$. This is a standard smoothness-type condition and is satisfied, for example, by losses with Lipschitz gradients or bounded curvature in directions orthogonal to the dominant subspace.

The second inequality formalizes that, within the subspace $V$, the loss at $v(y)$ is well approximated by a weighted combination of losses along the basis directions $\{v_i\}_{i=1}^k$, up to a controlled error governed by the parameter $\gamma$. Importantly, this condition does not require the loss to be additive, separable, or exactly quadratic in the coordinates $\alpha_i$. Interaction terms and higher-order effects are permitted and are absorbed into the $\gamma$ term. Such approximate decompositions arise naturally when the subspace $V$ captures the dominant directions of variation or curvature of the loss, for instance when $V$ consists of leading singular vectors, principal components, or basis directions from a low-rank factorization.

Overall, Assumption 2.1 quantifies the extent to which the loss function admits a low-dimensional algebraic approximation while allowing significantly worse approximation quality outside the subspace $V$. The regime in which $\lambda \gg \gamma$ corresponds to settings where directions orthogonal to $V$ are less informative or less well modeled, which is precisely the setting targeted by our approach.

These assumptions are natural in many machine learning settings. For example, in low-rank regression, PCA, or matrix completion, the dominant directions of the data capture most of the variance, while deviations along the orthogonal directions contribute minimally to the loss. In LLM fine-tuning or embedding-based models, top singular vectors often align with the most informative components, and residual directions carry less signal. Similar behavior is observed in low-rank adaptation techniques such as LoRA (Hu et al., 2022; Xu et al., 2024; Wu et al., 2024; Li et al., 2024), where trainable low-rank matrices capture the key directions in the parameter space. This indicates that many real-world

datasets are approximately low-rank, making these assumptions a reasonable abstraction for constructing coresets and selecting informative data efficiently. Our first main theoretical result is the following:

**Theorem 2.2** (Coreset Guarantee for Loss Approximation). *Let $D$ be a dataset of $n$ points with an embedding $E$, and suppose the loss function $\ell$ satisfies Assumption 2.1 with constants $\gamma, \lambda > 0$. Let*

$$\Phi_k(D) = \min_{\substack{D_k \in \mathbb{R}^{n \times m} \\ \mathrm{rank}(D_k) \leq k}} \|D - D_k\|_F^2$$

*denote the best rank-$k$ approximation cost of $D$. Then there exists a randomized algorithm that constructs a weighted subset $S \subseteq \mathbb{R}^m$ of size $s = \mathcal{O}\left(\frac{1}{\varepsilon^2}\right)$ with weights $w(x)$ such that, with probability at least $0.9$,*

$$\left| \sum_{x \in D} \ell(x) - \sum_{x \in S} w(x)\, \ell(x) \right| \tag{1}$$

$$\leq \varepsilon \cdot \left( \sum_{x \in D} \ell(x) + \gamma \|D\|_F^2 + \gamma k |D| \max \ell + 2\lambda \cdot \Phi_k(D) \right).$$

*Equivalently, the weighted average loss on $S$ is within a $(1 \pm \varepsilon)$ factor of the true average loss, up to an additive term proportional to $\Phi_k(D)/n$.*

Note that in Theorem 2.2, $\Phi_k(D) = \min_{\substack{D_k \in \mathbb{R}^{n \times m} \\ \mathrm{rank}(D_k) \leq k}} \|D - D_k\|_F^2$ denotes the squared Frobenius norm error of the best rank-$k$ approximation of the dataset $D$, which captures how well $D$ can be represented by $k$ components and will play a key role in our coreset guarantees. Thus, Theorem 2.2 formalizes the intuition that a small, carefully selected subset of data can effectively represent the loss of the entire dataset under low-rank structure assumptions. The theorem guarantees that a weighted subset $S$ of size $\mathcal{O}\left(\frac{1}{\varepsilon^2}\right)$ suffices to approximate the total loss over $D$ within a factor of $(1 \pm \varepsilon)$, up to an additive term proportional to $\Phi_k(D)$, the optimal rank-$k$ approximation error. The additive error in the theorem depends on $\Phi_k(D)$, the optimal low-rank approximation error. Datasets that are nearly low-rank yield smaller $\Phi_k(D)$, and hence the coreset more accurately preserves the total loss. This also indicates a tradeoff analogous to clustering: if the data contains significant outliers or high-rank noise, the bound increases, reflecting the inherent difficulty of representing such datasets with few points. Unlike clustering-based methods, however, the low-rank approach explicitly targets directions of high variance and information content, making it more robust in high-dimensional or unbalanced settings. Practically, this result implies that training or fine-tuning models on $S$ incurs minimal loss in accuracy while substantially reducing computational cost. Our experiments show that using subsets constructed via Theorem 2.2 achieves competitive or superior performance

compared to existing uniform or sensitivity sampling-based selection methods, providing a new practical sampling strategy for active regression (Chen & Price, 2019; Chen & Derezinski, 2021; Parulekar et al., 2021; Musco et al., 2022; Meyer et al., 2023; Woodruff & Yasuda, 2023).

On the other hand, one immediate concern regarding Theorem 2.2 is that the weighted set $S$ need not be a subset of the input dataset $D$. While the factors $S$ could potentially still be labeled by an external source, we also give the following guarantees based on the row subset selection problem:

**Theorem 2.3** (Coreset Guarantee for Loss Approximation via Row Subset Selection). *Let $D \subseteq \mathbb{R}^m$ be a dataset of $n$ points with an embedding $E$, and suppose the loss function $\ell$ satisfies Assumption 2.1 with constants $\gamma, \lambda > 0$. Let*

$$\Phi_k(D) = \min_{\substack{C \subseteq D \\ |C| = k}} \min_{A \in \mathbb{R}^{k \times m}} \|D - CA\|_F^2$$

*denote the optimal row subset selection cost using $k$ rows from $D$.*

*Then there exists a randomized algorithm that constructs a weighted subset $S \subseteq D$ of size $s = \mathcal{O}\left(\frac{1}{\varepsilon^2}\right)$ with weights $w(x)$ such that, with probability at least $0.9$,*

$$\left| \sum_{x \in D} \ell(x) - \sum_{x \in S} w(x)\, \ell(x) \right| \leq \tag{2}$$

$$\varepsilon \cdot \left( \sum_{x \in D} \ell(x) + \gamma \|D\|_F^2 + \gamma k |D| \max \ell + 2\lambda \cdot \Phi_k(D) \right).$$

*Equivalently, the weighted average loss over the selected rows $S$ approximates the true average loss over $D$ to within a $(1 \pm \varepsilon)$ factor, up to an additive term proportional to $\Phi_k(D)/n$.*

## 3. Problem Definition

### 3.1. Batch Data Selection and Loss Decomposition

We formally define the batch data selection problem in the context of low-rank losses. Let $D = \{(x_i, y_i)\}_{i=1}^n$ be a dataset sampled i.i.d. from a distribution $\mathcal{P}$ over $\mathcal{X} \times \mathcal{Y}$. We assume without loss of generality that $\mathcal{X} = \mathbb{R}^m$ and so we interchangeably use $D$ as the dataset and as a matrix of dimension $n \times m$. Given a sample $x$ and its label $y$, an algorithm $\mathcal{A}$ trains a model to produce a predicted label $\widehat{y}$, and incurs a loss based on the discrepancy between $y$ and $\widehat{y}$. We denote this loss by $\ell(x, y; \mathcal{A})$. The goal is to select a subset $S \subseteq D$ of size at most $s$ and associate a weight function $w : S \to \mathbb{R}^+$ such that

$$\Delta(S) := \left| \sum_{i=1}^n \ell(x_i, y_i; \mathcal{A}) - \sum_{x \in S} w(x)\, \ell(x, y; \mathcal{A}) \right|$$

is minimized, while keeping the number of expensive model evaluations (i.e., queries to $\ell$) small. Observe that the expected loss of $\mathcal{A}$ can be decomposed as

$$
\mathbb{E}_{(x,y)\sim\mathcal{P}}\ell(x,y;\mathcal{A})
$$

$$
\leq \underbrace{\left| \mathbb{E}_{(x,y)\sim\mathcal{P}}\ell(x,y;\mathcal{A}) - \frac{1}{n}\sum_{i=1}^{n}\ell(x_i,y_i;\mathcal{A}) \right|}_{\text{Generalization Error}}
$$

$$
+ \underbrace{\frac{1}{|C|}\sum_{j\in C}\ell(\widetilde{x}_j,\widetilde{y}_j;\mathcal{A})}_{\text{Training Error}}
$$

$$
+ \underbrace{\left| \frac{1}{n}\sum_{i=1}^{n}\ell(x_i,y_i;\mathcal{A}) - \frac{1}{|C|}\sum_{j\in C}\ell(x_j,y_j;\mathcal{A}) \right|}_{\text{Coreset Loss}},
$$

where $\widetilde{S} = \{(\widetilde{x}_j,\widetilde{y}_j)\}_{j\in C}$ is a coreset constructed from $S$ via some algorithm $\mathcal{A}$. This decomposition clarifies the sources of error in batch selection: the generalization error measures the gap between empirical and population loss, the training error captures how well the model fits the selected coreset, and the coreset loss quantifies how faithfully the coreset approximates the full dataset. Our low-rank sampling strategy explicitly targets minimizing the coreset loss while requiring only a small number of expensive evaluations of $\ell$, ensuring efficient and effective model training.

We remark that like (Axiotis et al., 2024), our formulation of data selection allows for *weighted sampling*, where each selected point can carry an individual weight $w(x)$, rather than assuming uniform weights like (Sener & Savarese, 2018). This is natural in the low-rank setting, where sampling probabilities derived from leverage scores or spectral sensitivities inherently produce non-uniform contributions to the coreset. Furthermore, rather than focusing on the loss after retraining the model on the subset, we consider the current model loss $\ell(x,y;\mathcal{A})$. Intuitively, if a subset $S$ approximates the loss of the full dataset well under the current model, it contains representative points that capture the dominant directions of the data. This allows subsequent training on $S$ to closely approximate training on the entire dataset without requiring assumptions on the label distribution or zero training loss. By framing the problem in this way, we can provide strong theoretical guarantees for low-rank coresets while maintaining flexibility and applicability to a wide range of models and loss functions.

### 3.2. Algorithm: Sensitivity Sampling for Low-Rank Loss Approximation

To efficiently construct the low-rank approximation coreset, we propose an algorithm that leverages sensitivity sampling based on the low-rank structure of $D$. Instead of clustering, we use low-rank approximation techniques (e.g., via sin-

gular value decomposition) to compute importance scores for data points. From the above assumptions, we show that a carefully selected small subset of data, constructed via sensitivity sampling based on low-rank structure, provides a provably accurate approximation of the overall loss. The proof of Theorem 2.2 leverages the low-rank structure of the dataset to construct an importance-weighted coreset. The key idea is to decompose each data point $x$ into two components: its projection $v(x)$ onto a low-rank subspace $V$, and its residual $r(x)$ orthogonal to $V$. The Lipschitz-like and basis decomposition assumptions ensure that the loss $\ell(x)$ can be tightly approximated by the contributions along the basis directions plus a small penalty for the residual. Intuitively, this means that the dominant directions of variance capture most of the loss, while the orthogonal directions contribute only a limited, controllable amount:

$$
\min_{\alpha}\sum_{j}p_{x_j}\ell\left(\sum_{i}\alpha_{j,i}^2 x_{j,i}\right)
$$

Using this decomposition, the algorithm defines a sensitivity score for each point, reflecting how much it contributes to the total loss relative to its projection and residual. Sampling points proportionally to these scores ensures that high-impact points are more likely to be included in the coreset. By weighting the sampled points appropriately, the resulting estimator becomes unbiased. A standard concentration inequality is then used to bound the deviation of the weighted sum from the total loss, giving the high-probability guarantee. Overall, the proof formalizes the intuition that a small, carefully weighted subset of points suffices to approximate the loss of the entire dataset, with an additive term proportional to the optimal rank-$k$ approximation error $\Phi_k(D)$.

We note that there exist many algorithms that achieve a bicriteria approximation for row subset selection:

**Theorem 3.1.** *(Guruswami & Sinop, 2012) There exists a polynomial-time algorithm that selects at most $2k$ columns that serve as a constant-factor approximation to the best rank-$k$ approximation.*

We defer the full proof of Theorem 2.3 to Appendix C. We also give the algorithm and analysis corresponding to Theorem 2.2, when the low-rank factors need not be data points to receive labels.

## 4. Real-World Dataset Experiments

In this section, we evaluate the effectiveness of our low-rank sensitivity sampling method on a variety of real-world datasets and downstream tasks. We begin with classical tabular datasets to measure coreset approximation quality and its impact on predictive performance. We then move on to large-scale language model fine-tuning experiments, where

**Algorithm 1** Sensitivity Sampling for Loss Approximation via Row Subset Selection

---

**Input:** Dataset $D = \{x_1, \ldots, x_n\} \subseteq \mathbb{R}^m$; target subset size $k$; error parameter $\varepsilon > 0$; constants $\lambda, \gamma$ corresponding to Assumption 2.1.

**Output:** A weighted subset $S \subseteq D$ of size $s$ that approximates the total loss.

1: Compute a row subset $C \subseteq D$ (e.g., via leverage-score or adaptive sampling)
2: Let $V = \text{span}(C)$ and let $v_1, \ldots, v_t$ be a basis for $V$
3: For each point $x \in D$, compute the residual vector

$$r(x) \leftarrow x - \text{Proj}(x, V).$$

4: Let $\text{Proj}(x, V) = \alpha_1 v_1 + \ldots + \alpha_t v_t$
5: $\sigma(x) \leftarrow (\gamma + 1)(\alpha_1^2 \ell(v_1) + \ldots + \alpha_t^2 \ell(v_t)) + \gamma k \xi + \lambda \|r(x)\|_2^2,$
6: Normalize the scores to obtain a probability distribution:

$$p(x) = \frac{\sigma(x)}{\sum_{y \in D} \sigma(y)}.$$

7: Set $s \leftarrow \left\lceil \varepsilon^{-2} \left( 2 + \frac{2\varepsilon}{3} \right) \right\rceil.$
8: Sample $s$ points from $D$ independently according to $\{p(x)\}_{x \in D}.$
9: **for** each sampled point $x$ **do**
10:     Set its weight $w(x) \leftarrow \frac{1}{s\, p(x)}.$
11: **end for**
12: **return** $S$ with associated weight function $w(\cdot).$

---

data selection plays a crucial role in reducing computational cost while maintaining or improving accuracy. Across all experiments, we compare our approach against uniform sampling, clustering-based methods, and other state-of-the-art sensitivity sampling techniques to highlight the advantages of leveraging low-rank structure in the data.

### 4.1. Credit card dataset

We evaluate on the Default of Credit Card Clients dataset (Yeh, 2016; Yeh & Lien, 2009), which contains $30\,000$ records described by 23 attributes, including six months of previous bill statements, repayment statuses, credit limits and demographic variables. The binary label represents default on the next month's payment (22% positive rate). This heterogeneous, imbalanced dataset is a standard benchmark for subsampling and downstream classification. We empirically verify Assumption 2.1 on this task; see Appendix D for details.

**Experimental setup.** All coreset experiments begin by loading the full dataset, renaming the column `default payment next month` to `Class`, dropping the `ID`, and applying z-score normalization to all 23 feature columns.

We then compute the per-point squared norms $\ell_i = \|x_i\|_2^2$ and the true sum $L_{\text{true}} = \sum_{i=1}^n \ell_i$. We vary coreset size $s \in \{1000, 2000, 3000, 4000, 5000\}$ and repeat 100 independent trials of each method to average results.

For random sampling we draw $s$ points *with replacement* uniformly at random and assign each weight $n/s$. For clustering, we run a K-Means++ algorithm with maximum 300 iterations. We repeat the clustering 10 times and pick the best results. on the standardized data, select the nearest training point to each centroid, and weight it by its cluster size. For sensitivity sampling (Algorithm 2), we run `TruncatedSVD(n_components=5)` from scikit-learn (Pedregosa et al., 2011) to obtain projection vectors, compute projected-loss term $(\gamma + 1)\,\alpha^2 \ell(v_i)$, basis-loss term $\gamma$, $k, \xi$, and residual-loss term $\lambda \|r_i\|^2$ with parameters $\gamma = 5$, $\lambda = 1$, smoothing $10^{-6}$, normalize to probabilities $p_i$, sample $s$ points *with replacement* according to $p_i$, and assign weights $1/(s\, p_i)$.

In the coreset-error experiment (Figure 1a) we measure error $= \left| \sum_j w_j \|x_j\|_2^2 - L_{\text{true}} \right|$, for each trial and average across trials. In the downstream accuracy experiment (Figure 1b) we first fit a full logistic regression model (`solver='liblinear'`, `class_weight='balanced'`, `max_iter=1000`) on the training set to obtain per-point logistic losses for sensitivity sampling, then for each $s$ and each method train a logistic model with identical hyperparameters on the weighted coreset and evaluate test accuracy on the held-out 20%.

**Results and discussion.** Figure 1a shows that sensitivity sampling results in the lowest approximation error at every sample size, reducing error by an order of magnitude relative to random sampling and by roughly 50% compared to clustering at $s = 1000$, with all methods converging as $s$ increases. Figure 1b then shows that logistic regression trained on sensitivity coresets attains up to 74% test accuracy at $s = 5000$, clustering coresets reach around 70%, and random sampling only about 67%. These results confirm that the low-rank sensitivity algorithm not only tightens coreset-error bounds but also translates into improved predictive performance on an imbalanced, real-world financial dataset.

### 4.2. LLM Fine-tuning Experiments

#### 4.2.1. SETTING

**Models and datasets.** We fine-tune the standard instruction-tuned Llama3-8B (Dubey et al., 2024) and Qwen2.5-3B (Yang et al., 2024) models on three challenging downstream datasets: Grade-School Math (GSM8k) (Cobbe et al., 2021) with 7.47k training and 1.32k test samples, ViGGO (Juraska et al., 2019) with 5.1k training and 1.08k test samples, and

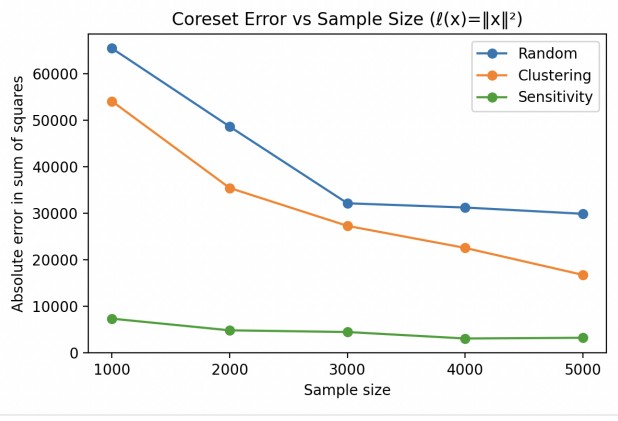
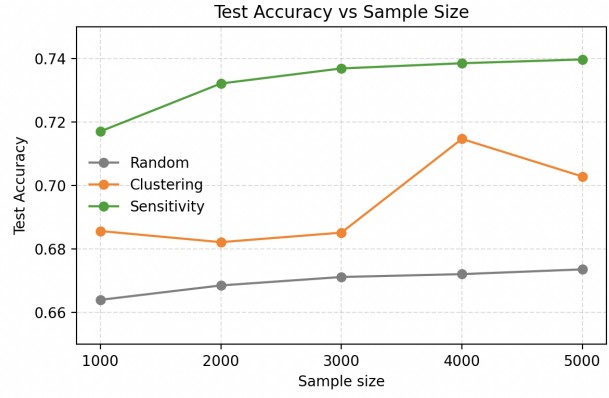

*(a)* Absolute error in sum of squares vs. sample size.    *(b)* Test accuracy vs. sample size.

*Figure 1.* Comparison of Random, Clustering, and Low-Rank Sensitivity sampling on the Default Credit Card dataset.

SQL generation (Yu et al., 2018; Zhong et al., 2017) with 30k training and 1k test samples. This fine-tuning setup is widely used and has appeared in multiple research papers (Ashkboos et al., 2025; Chen et al., 2025; Nikdan et al., 2024). These tasks are specifically selected because the base models perform poorly on them, making them well-suited for fine-tuning. We closely follow the evaluation strategy of (Ashkboos et al., 2025).

**Hyperparameters.** We largely adopt the training hyperparameters from the HALO code base (Ashkboos et al., 2025). For fine-tuning Llama3-8B-Instruct, we use the Adam optimizer for one epoch with learning rates $6 \times 10^{-6}$, $4 \times 10^{-5}$, and $3 \times 10^{-5}$ for GSM8k, ViGGO, and SQL, respectively. All dataset samples are encoded using the standard and efficient BERT embeddings (Devlin et al., 2019b). For clustering, we employ k-means++ and map each centroid to its closest sample in the dataset. The clustering procedure is repeated 10 times, and the best results are retained. For landmark selection in low-rank approximation, we employ leverage score sampling. By default, the number of clusters/landmarks is set to 20% of the total number of available samples, following the experimental setting of (Axiotis et al., 2024). Regarding the parameters of Assumption 2.1, we tune the $\lambda$ value and pick the top performing one when applicable. Additionally, we set $\gamma = 0$, and compute $\alpha$ values in the embeddings space using Kernel Ridge Regression (KRR) with an RBF kernel to find the linear combination of landmark loss values. For Qwen2.5-3B, we use the same hyperparameters except the learning rate on the ViGGO dataset, which is increased to $10^{-4}$

**Baselines.** We consider five baselines: 1) *Full training*: where the data selection is skipped and the model is trained on the full dataset, 2) *Uniform sampling*, where the subset samples are selected uniformly at random, 3) K-Center, the method from (Sener & Savarese, 2017), 4) *Clustering-based*

*sensitivity sampling* (Axiotis et al., 2024), which similar to our method, uses sensitivity sampling, but relies on clustering rather than low-rank approximation, and 5) Graph Cut, which maximizes the sum of similarities between selected and unselected samples, implemented in (Schreiber et al., 2020).

### 4.2.2. RESULTS

**Main results.** We begin by fine-tuning the Llama3-8B model on 25%, 12.5%, and 6.25% of each dataset, selected using various sampling methods. Table 1 reports the validation accuracy of our method compared to the baselines. The results show that our method consistently outperforms uniform sampling. On average, it also achieves higher accuracy than clustering-based sensitivity sampling (Axiotis et al., 2024) in most datasets, demonstrating the benefit of leveraging low-rank approximation for data selection.

**Runtime discussion.** The selection process for both cluster-based and low-rank sensitivity sampling requires forward passes on $k = 20\%$ of the dataset. Assuming a backward pass is twice as expensive as a forward pass (Kaplan et al., 2020), this corresponds to approximately 6.67% of the total runtime for training on the full dataset.

**Study on dataset structure.** Here we analyze the training split of GSM8k to examine whether it exhibits a more clustered or low-rank structure, such as Assumption 2.1 for the latter. To this end, across a range of $k$ values, we perform the following experiments:

$i)$ We cluster the per-sample embeddings into $k$ clusters and compute the average euclidean distance from each sample to its closest cluster center, and compare this with the average low-rank approximation error when representing the dataset using $k$ basis samples.

$ii)$ We measure the average loss difference between each

*Table 1.* End-to-end fine-tuning validation accuracy on different baselines and datasets. BERT embeddings are used and $k$ is fixed to 25% of the dataset. SS stands for Sensitivity Sampling.

| Dataset | GSM8k | | | ViGGO | | | SQL | | | Average | | |
|---|---|---|---|---|---|---|---|---|---|---|---|---|
| Sampling Ratio | 25% | 12.5% | 6.25% | 25% | 12.5% | 6.25% | 25% | 12.5% | 6.25% | 25% | 12.5% | 6.25% |
| *Uniform Sampling* | $67.7 \pm 0.3$ | $65.3 \pm 0.2$ | $63.5 \pm 0.5$ | $86.3 \pm 0.7$ | $68.3 \pm 4.1$ | $26.2 \pm 6.2$ | $75.6 \pm 0.5$ | $74.1 \pm 0.5$ | $66.2 \pm 3.5$ | 76.5 | 69.2 | 52.0 |
| *K-Center* | $67.4 \pm 0.3$ | $65.3 \pm 0.5$ | $58.4 \pm 3.3$ | $80.3 \pm 2.5$ | $51.0 \pm 1.8$ | $25.5 \pm 7.7$ | $\mathbf{77.1 \pm 0.3}$ | $\mathbf{75.3 \pm 0.4}$ | $\mathbf{73.0 \pm 0.5}$ | 74.9 | 63.9 | 52.3 |
| *Graph Cut* | $67.0 \pm 0.5$ | $\mathbf{67.7 \pm 0.7}$ | $64.9 \pm 1.6$ | $81.7 \pm 2.9$ | $58.6 \pm 3.8$ | $13.6 \pm 4.7$ | $73.3 \pm 0.6$ | $71.1 \pm 0.2$ | $58.5 \pm 7.5$ | 74.0 | 65.8 | 45.7 |
| *Clustering-based SS* | $\mathbf{70.2 \pm 0.1}$ | $66.6 \pm 1.2$ | $65.2 \pm 1.1$ | $86.6 \pm 2.8$ | $\mathbf{72.8 \pm 1.7}$ | $\mathbf{30.3 \pm 3.9}$ | $75.6 \pm 0.5$ | $73.7 \pm 0.5$ | $68.3 \pm 3.6$ | 77.5 | $\mathbf{71.0}$ | 54.6 |
| *Low-rank SS (ours)* | $68.4 \pm 0.1$ | $67.1 \pm 0.9$ | $\mathbf{65.4 \pm 1.6}$ | $\mathbf{88.3 \pm 0.2}$ | $69.7 \pm 5.2$ | $28.8 \pm 1.1$ | $76.1 \pm 0.2$ | $74.4 \pm 0.2$ | $70.4 \pm 1.0$ | $\mathbf{77.6}$ | 70.4 | $\mathbf{54.9}$ |
| *Full (100%)* | | $69.3 \pm 0.5$ | | | $94.0 \pm 0.3$ | | | $79.9 \pm 0.5$ | | | 81.1 | |

sample's true loss and that of its nearest cluster center, and compare it against the average difference between the true loss and the low-rank approximated loss.

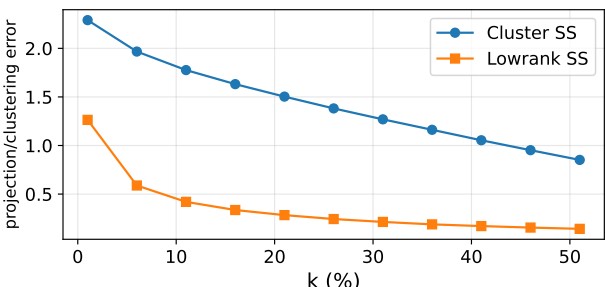

*(a)* Average projection/clustering error

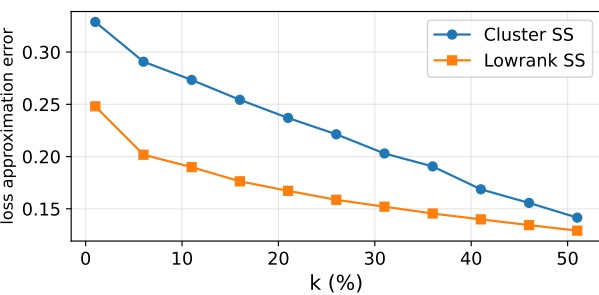

*(b)* Average loss approximation error

*Figure 2.* Comparison of cluster-based and low-rank sensitivity sampling methods on the GSM8k dataset using BERT embeddings. The values of $k$ are expressed as percentages of the entire dataset.

Figure 2 shows that, in both cases, the dataset yields a smaller error under the low-rank approximation, supporting Assumption 2.1 and our claim that the dataset (GSM8k in this case) is more aligned with a low-rank structure than with a purely clustered one.

**Average loss approximation quality.** We next investigate how well the (weighted) subset selected by each method approximates the average loss. We vary $k$ and $\lambda$, and in each case, select 2000 samples from the GSM8k dataset and measure the average loss approximation error ($\Delta(S)$, Section 3.1). Figure 3 presents heatmaps for both cluster-based and low-rank sensitivity sampling, showing that low-rank

consistently achieves lower error. An interesting observation in the clustering case is that, at $\lambda = 1$, increasing the number of clusters degrades the approximation quality. This occurs because a large $\lambda$ causes the sampling score to be dominated by the geometric distance $r$, leading the algorithm to prioritize outliers over points from high-loss regions. When the number of clusters $k$ increases, the data space is partitioned more finely, reducing $r$ for inlier points and further biasing the selection toward outliers. Consequently, the selected subset becomes less representative of the overall distribution, resulting in poorer average loss approximation. A similar effect occurs for low-rank sampling at $\lambda \geq 100$, though these cases are omitted from the plots for clarity.

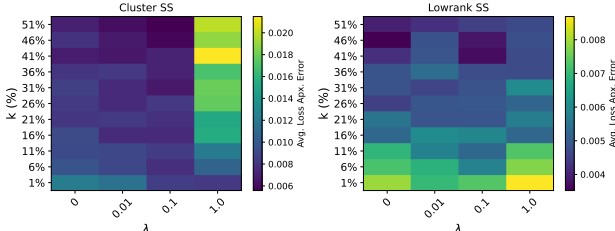

*Figure 3.* Average loss approximation error across different $k$ and $\lambda$ values. In each case, 2000 (weighted) samples are selected from the GSM8k dataset, and the average of 100 trials is reported.

**Alternative objective and embedding.** Following (Axiotis et al., 2024), we repeat our 12.5% selection experiments on GSM8k, but replace the loss with the norm of per-sample gradients in Algorithm 2. Gradient norm serves as a proxy for capturing training dynamics (Axiotis et al., 2024). Figure 4 compares cluster-based and low-rank sensitivity sampling across different $\lambda$ values. The results indicate a slight advantage for low-rank sampling, which also appears more robust to the choice of $\lambda$, consistently outperforming uniform sampling for all values considered. Additionally, the same figure presents results for replacing BERT embeddings (Devlin et al., 2019b) with GTR-base embeddings (Ni et al., 2021). The results indicate that our positive findings remain consistent with these embeddings as well.

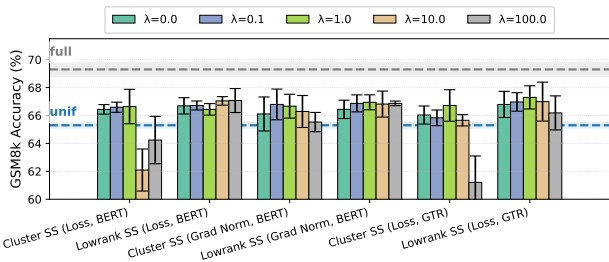

*Figure 4.* Comparison of alternative objective functions (loss vs. gradient norm) and embedding functions, including BERT (Devlin et al., 2019b) and GTR (Ni et al., 2021), in terms of end-to-end validation accuracy, across various $\lambda$ choices. Experiments are conducted on the GSM8k dataset with $k$ fixed at $20\%$, and the selected subset size fixed at $12.5\%$ of the dataset.

*Table 2.* ViGGO results with Qwen2.5-3B at $25\%$ and $12.5\%$ compression. Our method outperforms all baselines at both compression rates.

| Method | 25% | 12.5% |
|---|---|---|
| Uniform | 68.35 | 45.38 |
| K-Center | 59.24 | 43.14 |
| Graph Cut | 62.89 | 38.94 |
| Clustering-based SS | 71.57 | 48.46 |
| Lowrank SS (ours) | **75.91** | **51.54** |

*Table 3.* Transferability of selected subsets on ViGGO: subsets are selected with either Qwen2.5-3B or Llama3-8B and used to train Qwen2.5-3B. Subsets selected by the larger Llama3-8B transfer well and even surpass selection with the target model itself.

| Method | Selector | 25% | 12.5% |
|---|---|---|---|
| Uniform | – | 68.35 | 45.38 |
| K-Center | – | 59.24 | 43.14 |
| Clustering-based SS | Qwen | 71.57 | 48.46 |
| Lowrank SS (ours) | Qwen | 75.91 | 51.54 |
| Clustering-based SS | Llama | 72.27 | 47.06 |
| Lowrank SS (ours) | Llama | **76.05** | **55.74** |

*Table 4.* Ablation on the basis-selection algorithm within Lowrank SS, evaluated on ViGGO at $12.5\%$ compression with Qwen2.5-3B. Leverage score sampling performs best, while $k$-medians lags both alternatives by a clear margin.

| Basis Selection | Accuracy |
|---|---|
| Uniform | 50.70 |
| $k$-Medians | 45.80 |
| Leverage Score (ours) | **51.54** |

**Qwen2.5-3B Results.** To verify that our method generalizes beyond a single model, we repeat the ViGGO experiments on Qwen2.5-3B (Yang et al., 2024) at $25\%$ and $12.5\%$ compression rates. As summarized in Table 2, Lowrank SS continues to outperform all baselines by clear margins, improving over the strongest baseline (Clustering-based SS) by $4.34$ and $3.08$ points at $25\%$ and $12.5\%$ compression, respectively.

**Transferability.** We ask whether subsets selected by one model transfer to another. Specifically, we take the ViGGO $25\%$ and $12.5\%$ subsets selected using Llama3-8B and use them to train Qwen2.5-3B. Table 3 reports these results. We find that the selected subsets transfer remarkably well: subsets chosen with Llama3-8B not only match but *exceed* those chosen with Qwen2.5-3B itself when used to train Qwen2.5-3B, with the gap most pronounced at $12.5\%$ compression ($55.74$ vs. $51.54$). This suggests that the signal captured by our low-rank criterion reflects intrinsic properties of the data rather than model-specific artifacts.

**Effect of Basis Selection Algorithm.** A natural question is how sensitive our method is to the specific algorithm used to choose the basis samples. To isolate this factor, we repeat the ViGGO $12.5\%$ experiments on Qwen2.5-3B while varying only the basis-selection procedure: (1) uniform random selection of basis samples, (2) selecting basis samples via $k$-medians clustering, and (3) leverage score

sampling, as used in our main method. Table 4 reports the resulting accuracies. Leverage score sampling yields the best performance, with a small edge over uniform basis selection ($51.54$ vs. $50.70$). Both, however, outperform $k$-medians by a notable margin ($\sim$ 5–6 points), suggesting that representativeness-based clustering is a poor fit for choosing a basis in our low-rank framework. Notably, even the uniform variant remains competitive with the strongest baselines in Table 2, indicating that the gains of our method come primarily from the low-rank selection criterion itself, with the basis-sampling strategy providing an additional, smaller improvement.

## Acknowledgments

David P. Woodruff is supported in part Office of Naval Research award number N000142112647, and a Simons Investigator Award. Samson Zhou is supported in part by NSF CCF-2335411. Samson Zhou gratefully acknowledges funding provided by the Oak Ridge Associated Universities (ORAU) Ralph E. Powe Junior Faculty Enhancement Award.

## Impact Statement

This paper presents work whose goal is to advance the field of machine learning. There are many potential societal consequences of our work, none of which we feel must be specifically highlighted here.

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

# A. Related Works

Deep learning methods have become the standard for large-scale image classification and related tasks. While these models achieve state-of-the-art performance, they typically require very large labeled datasets to train effectively. This dependency on large-scale supervision motivates the study of techniques that can reduce the labeling burden without sacrificing performance, such as active learning and subset selection.

**Active learning.** A large body of work has examined the theoretical underpinnings of active learning. Classical results show that greedy selection is impossible in a fully agnostic setting (Dasgupta, 2004), yet refined analyses demonstrate stronger guarantees under assumptions such as realizability (Gonen et al., 2013) or bounded disagreement coefficients (Hanneke, 2007). Other approaches justify greedy strategies in the batch setting via importance sampling (Ganti & Gray, 2012). Although these works provide rigorous guarantees, they do not address the large-scale deep learning problems that motivate our study.

Complementing these theoretical contributions, several algorithms have been designed specifically for CNNs. (Wang et al., 2017) propose auto-labeling of confident predictions while querying uncertain points, and (Stark et al., 2015) develop a method tailored for CAPTCHA recognition. These CNN-oriented techniques succeed in narrow domains but do not scale to general-purpose image classification tasks.

A different stream of research views active learning through the lens of optimization. Formulations that balance uncertainty and diversity often cast the problem as a discrete program with convex relaxations (Elhamifar et al., 2013; Yang et al., 2015; Guo, 2010), but these require $n^2$ variables for $n$ data points, making them impractical for large datasets. More specialized efforts adapt active learning to k-nearest neighbors, naive Bayes, or logistic regression (Wei et al., 2015; Hoi et al., 2006; Guo & Schuurmans, 2007; Yu et al., 2006). Within this optimization-oriented family, (Demir et al., 2011) propose a two-stage approach that first filters uncertain points and then enforces diversity. Other approaches include efforts are by (Joshi et al., 2010) and (Wang & Ye, 2015): the former introduces a related optimization problem without theory, while the latter minimizes maximum mean discrepancy. Our framework builds on these ideas by introducing the notion of core-set loss, providing both a theoretical foundation and practical applicability to deep models.

Finally, classical acquisition strategies remain an influential part of the literature. Early surveys such as (Settles, 2009) summarize information-theoretic approaches (MacKay, 1992), ensemble-based methods (McCallum & Nigam, 1998; Freund et al., 1997), and uncertainty-driven heuristics (Tong & Koller, 2001a; Joshi et al., 2009; Li & Guo, 2013). In particular, uncertainty-based sampling focuses on querying ambiguous points, using entropy (Joshi et al., 2009) or margin-based distances (Tong & Koller, 2001a; Brinker, 2003). Bayesian active learning has also been widely studied, traditionally with Gaussian processes to estimate error reduction or predictive improvement (Roy & McCallum, 2001; Kapoor et al., 2007). While powerful in small-scale settings, these approaches do not scale to modern applications.

# B. Preliminaries

Let $V = \operatorname{span}\{v_1, \ldots, v_k\} \subset \mathbb{R}^d$ be a $k$-dimensional subspace with an orthonormal basis $\{v_1, \ldots, v_k\}$. For the standard inner product $\langle x, v_i \rangle$, the projection of a vector $x \in \mathbb{R}^d$ onto $V$ is defined as $\operatorname{Proj}(x, V) := \sum_{i=1}^{k} \langle x, v_i \rangle v_i$. Let $A \in \mathbb{R}^{n \times d}$ be a data matrix. The *singular value decomposition (SVD)* of $A$ is $A = U \Sigma V^{\top}$, where $U \in \mathbb{R}^{n \times n}$ and $V \in \mathbb{R}^{d \times d}$ are orthogonal matrices containing the left and right singular vectors, respectively, and $\Sigma \in \mathbb{R}^{n \times d}$ is a diagonal matrix with non-negative singular values $\sigma_1 \geq \sigma_2 \geq \ldots \geq 0$.

For a target rank $k < \min(n, d)$, the *best rank-$k$ approximation* of $A$ in the Frobenius norm is obtained by truncating the SVD to the top $k$ singular values $A_k = U_k \Sigma_k V_k^{\top}$, where $U_k \in \mathbb{R}^{n \times k}$, $V_k \in \mathbb{R}^{d \times k}$, and $\Sigma_k \in \mathbb{R}^{k \times k}$ contain the top $k$ singular vectors and singular values. The Eckart-Young-Mirsky theorem (Eckart & Young, 1936) guarantees that $A_k = \arg\min_{\substack{B \in \mathbb{R}^{n \times d} \\ \operatorname{rank}(B) \leq k}} \|A - B\|_F^2$, i.e., $A_k$ is the unique rank-$k$ matrix that minimizes the squared Frobenius norm of the approximation error. The optimal cost is often denoted by $\Phi_k(A) := \min_{\substack{B \in \mathbb{R}^{n \times d} \\ \operatorname{rank}(B) \leq k}} \|A - B\|_F^2 = \sum_{i=k+1}^{\min(n,d)} \sigma_i^2$. This low-rank approximation captures the most significant directions of variance in the data, and forms the foundation for coresets and data selection under low-rank losses.

# C. Missing Proofs

To derive theoretical guarantees for our low-rank sampling framework, we assume the following smoothness condition on the loss function $\ell$ and the low-dimensional factor $V$. Let $V = \mathrm{span}\{v_1, \ldots, v_k\}$ be a $k$-dimensional subspace, for instance corresponding to the top singular vectors, principal components, or basis directions from a low-rank factorization. For any point $y \in \mathbb{R}^d$, decompose

$$y = \alpha_1 v_1 + \cdots + \alpha_k v_k + r(y), \quad r(y) = \mathrm{Proj}_{V^\perp}(y),$$

where $\alpha_i = \langle y, v_i \rangle$ and $r(y)$ is the component orthogonal to $V$, i.e., the projection of $y$ onto $V^\perp$. Let $v(y) = \mathrm{Proj}_V(y) = \alpha_1 v_1 + \cdots + \alpha_k v_k$.

**Assumption C.1.** *We assume there exist constants $\lambda, \gamma > 0$ such that*

$$|\ell(y) - \ell(v(y))| \le \lambda \|r(y)\|_2^2, \qquad |\ell(v(y)) - (\alpha_1^2 \ell(v_1) + \cdots + \alpha_k^2 \ell(v_k))| \le \gamma \sum_{i=1}^k |\alpha_i^2 - 1| \, \ell(v_i).$$

Intuitively, this condition decomposes the loss at $y$ into two components: a weighted sum of the losses along each basis direction $v_i$, with weights $\alpha_i^2$, and a penalty proportional to the squared norm of the component orthogonal to $V$, $\|r(y)\|_2^2$.

---

**Algorithm 2** Sensitivity Sampling for Low-Rank Loss Approximation

---

**Input:** Dataset $D = \{x_1, \ldots, x_n\}$; target rank $k$; error parameter $\varepsilon > 0$; Constants $\lambda, \gamma$ corresponding to Assumption 2.1.
**Output:** A weighted subset $S \subseteq D$ of size $s$ that approximates the total loss.
1: Compute a rank-$k$ approximation $V$ of $D$ (e.g., via SVD) and let $v_1, \ldots, v_k$ be a basis for $V$
2: For each point $x \in D$, compute the residual vector

$$r(x) \leftarrow x - \mathrm{Proj}(x, V).$$

3: Let $\mathrm{Proj}(x, V) = \alpha_1 v_1 + \ldots + \alpha_k v_k$
4: $\sigma(x) \leftarrow (\gamma + 1)(\alpha_1^2 \ell(v_1) + \ldots + \alpha_k^2 \ell(v_k)) + \gamma k \xi + \lambda \|r(x)\|_2^2$,
5: Normalize the scores to obtain a probability distribution:

$$p(x) = \frac{\sigma(x)}{\sum_{y \in D} \sigma(y)}.$$

6: Set $s \leftarrow \left\lceil \varepsilon^{-2} \left( 2 + \frac{2\varepsilon}{3} \right) \right\rceil$.
7: Sample $s$ points from $D$ independently according to $\{p(x)\}_{x \in D}$.
8: **for** each sampled point $x$ **do**
9:     Set its weight $w(x) \leftarrow \frac{1}{s \, p(x)}$.
10: **end for**
11: **Return** $S$ with associated weight function $w(\cdot)$.

---

These assumptions are natural in many machine learning settings. For example, in low-rank regression, PCA, or matrix completion, the dominant directions of the data capture most of the variance, while deviations along the orthogonal directions contribute minimally to the loss. In LLM fine-tuning or embedding-based models, top singular vectors often align with the most informative components, and residual directions carry less signal. Similar behavior is observed in low-rank adaptation techniques such as LoRA (Hu et al., 2022; Xu et al., 2024; Wu et al., 2024; Li et al., 2024), where trainable low-rank matrices capture the key directions in the parameter space. This indicates that many real-world datasets are approximately low-rank, making these assumptions a reasonable abstraction for constructing coresets and selecting informative data efficiently.

Given Assumption C.1, our procedure in Algorithm 2 uses a form of importance sampling (Feldman & Langberg, 2011; Langberg & Schulman, 2010; Braverman et al., 2021a), to achieve loss proportional to the best possible loss achievable by low-rank approximation. We remark this is a standard procedure for constructing coresets (Har-Peled & Mazumdar, 2004; Chen, 2009; Feldman & Langberg, 2011; Langberg & Schulman, 2010; Phillips, 2016; Mirzasoleiman et al., 2020; Feldman, 2020; Braverman et al., 2021b; Mahabadi et al., 2022; Tukan et al., 2022; Mussay et al., 2022; Cohen-Addad et al., 2023; Jiang et al., 2023; Tukan et al., 2023; Zheng et al., 2023; Song et al., 2024; Kenneth-Mordoch & Sapir, 2025). Then our result is as follows:

**Theorem 2.2** (Coreset Guarantee for Loss Approximation). *Let $D$ be a dataset of $n$ points with an embedding $E$, and suppose the loss function $\ell$ satisfies Assumption 2.1 with constants $\gamma, \lambda > 0$. Let*

$$\Phi_k(D) = \min_{\substack{D_k \in \mathbb{R}^{n \times m} \\ \text{rank}(D_k) \leq k}} \|D - D_k\|_F^2$$

*denote the best rank-$k$ approximation cost of $D$. Then there exists a randomized algorithm that constructs a weighted subset $S \subseteq \mathbb{R}^m$ of size $s = \mathcal{O}\left(\frac{1}{\varepsilon^2}\right)$ with weights $w(x)$ such that, with probability at least $0.9$,*

$$\left| \sum_{x \in D} \ell(x) - \sum_{x \in S} w(x)\, \ell(x) \right| \tag{1}$$

$$\leq \varepsilon \cdot \left( \sum_{x \in D} \ell(x) + \gamma \|D\|_F^2 + \gamma k |D| \max \ell + 2\lambda \cdot \Phi_k(D) \right).$$

*Equivalently, the weighted average loss on $S$ is within a $(1 \pm \varepsilon)$ factor of the true average loss, up to an additive term proportional to $\Phi_k(D)/n$.*

*Proof.* Let $L := \sum_{x \in D} \ell(x)$ be the total loss over the dataset $D$, and define

$$\Phi_k(D) = \min_{\text{rank}(V) \leq k} \|D - V\|_F^2,$$

the best rank-$k$ approximation error of $D$. For every point $x \in D$, let $v(x) = \text{Proj}(x, V)$ be the projection of $x$ onto the chosen low-rank approximation $V$ and let $r(x) = x - \text{Proj}(x, V)$ be the orthogonal complement so that $x = v(x) + r(x)$. By the Lipschitz condition (with constant $\lambda$), we have for every $x \in D$:

$$\left| \ell(x) - \ell(v(x)) \right| \leq \lambda \cdot \|r(x)\|_2^2.$$

Suppose we have $v(x) = \alpha_1 v_1 + \ldots + \alpha_k v_k$. Then we also have

$$\left| \ell(v(x)) - (\alpha_1^2 \ell(v_1) + \ldots + \alpha_k^2 \ell(v_k)) \right| \leq \gamma \left( |\alpha_1^2 - 1| \ell(v_1) + \ldots + |\alpha_k^2 - 1| \ell(v_k) \right),$$

where $v(x) = \alpha_1 v_1 + \ldots + \alpha_k v_k$. Hence by triangle inequality, we have

$$\ell(x) \leq (\alpha_1^2 \ell(v_1) + \ldots + \alpha_k^2 \ell(v_k)) + \gamma \left( |\alpha_1^2 - 1| \ell(v_1) + \ldots + |\alpha_k^2 - 1| \ell(v_k) \right) + \lambda \|r(x)\|_2^2$$

$$\leq (\gamma + 1)(\alpha_1^2 \ell(v_1) + \ldots + \alpha_k^2 \ell(v_k)) + \gamma k \cdot \max_k \ell(v_k) + \lambda \|r(x)\|_2^2$$

and

$$(\alpha_1^2 \ell(v_1) + \ldots + \alpha_k^2 \ell(v_k)) \leq \ell(x) + \gamma \left( |\alpha_1^2 - 1| \ell(v_1) + \ldots + |\alpha_k^2 - 1| \ell(v_k) \right) + \lambda \|r(x)\|_2^2$$

$$\leq \ell(x) + \gamma (\|x\|_2^2 + k) \cdot \max_k \ell(v_k) + \lambda \|r(x)\|_2^2.$$

Let $\xi \geq \max_k \ell(v_k)$. Then we next define the sensitivity score for each $x \in D$ as the following:

$$\sigma(x) := (\gamma + 1)(\alpha_1^2 \ell(v_1) + \ldots + \alpha_k^2 \ell(v_k)) + \gamma k \xi + \lambda \|r(x)\|_2^2,$$

where $v(x) = \alpha_1 v_1 + \ldots + \alpha_k v_k$. Assign the sampling probability by normalizing these scores:

$$p(x) := \frac{\sigma(x)}{T}, \quad \text{where} \quad T := \sum_{y \in D} \sigma(y).$$

We now select $s$ independent samples (with replacement) from $D$ according to $p(x)$; define $S = \{x_1, \ldots, x_s\}$ as the resulting multiset. For every sample $x \in S$, define its weight as

$$w(x) := \frac{1}{s\, p(x)}.$$

Hence, the weighted loss estimator is

$$Z := \sum_{x \in S} w(x)\,\ell(x).$$

Through the linearity of expectation,

$$\mathbb{E}\big[\ell(x)w(x)\big] = \sum_{x \in D} p(x) \cdot \frac{\ell(x)}{s\,p(x)} = \frac{1}{s}\sum_{x \in D} \ell(x) = \frac{L}{s},$$

so $\mathbb{E}[Z] = L$; that is, the estimator is unbiased.

For a single sample, let

$$X = \ell(x)w(x) = \frac{\ell(x)}{s\,p(x)}.$$

Then its second moment is

$$\mathbb{E}[X^2] = \sum_{x \in D} p(x)\left(\frac{\ell(x)}{s\,p(x)}\right)^2 = \frac{1}{s^2}\sum_{x \in D}\frac{\ell(x)^2}{p(x)}.$$

Substituting $p(x) = \sigma(x)/T$, we get the following

$$\mathbb{E}[X^2] = \frac{T}{s^2}\sum_{x \in D}\frac{\ell(x)^2}{(\gamma+1)(\alpha_1^2\ell(v_1) + \ldots + \alpha_k^2\ell(v_k)) + \gamma k\xi + \lambda\|r(x)\|_2^2}.$$

Since $\ell(x) \le (\gamma+1)(\alpha_1^2\ell(v_1) + \ldots + \alpha_k^2\ell(v_k)) + \gamma k\xi) + \lambda\|r(x)\|_2^2$, it follows that

$$\frac{\ell(x)^2}{(\gamma+1)(\alpha_1^2\ell(v_1) + \ldots + \alpha_k^2\ell(v_k)) + \gamma k\xi + \lambda\|r(x)\|_2^2} \le \ell(x).$$

Thus,

$$\mathbb{E}[X^2] \le \frac{T}{s^2}\sum_{x \in D}\ell(x) = \frac{L\,T}{s^2}.$$

Summing over all $s$ samples, we have

$$\sum_{i=1}^{s}\mathbb{E}[X_i^2] \le \frac{L\,T}{s}.$$

Using the bound $T \le L + \gamma(\|D\|_F^2 + k|D|)\xi + \lambda R$, where

$$R := \sum_{x \in D}\|r(x)\|_2^2,$$

we obtain

$$\sum_{i=1}^{s}\mathbb{E}[X_i^2] \le \frac{\big(L + \gamma(\|D\|_F^2 + k|D|)\xi + \lambda R\big)^2}{s}.$$

For any point $x \in D$, its weighted contribution is

$$\ell(x)w(x) = \frac{\ell(x)}{s}\frac{1}{p(x)} = \frac{\ell(x)}{s}\frac{T}{\sigma(x)}.$$

Since $\ell(x) \le \sigma(x)$, it follows that

$$\ell(x)w(x) \le \frac{T}{s} \le \frac{L + \gamma(\|D\|_F^2 + k|D|)\xi + \lambda R}{s}.$$

Thus, if we set

$$M := \frac{L + \gamma(\|D\|_F^2 + k|D|)\xi + \lambda R}{s},$$

then $|X_i| \leq M$ for every sample.

Let

$$Z = \sum_{i=1}^{s} X_i = \sum_{x \in S} w(x)\, \ell(x).$$

By Bernstein's inequality, for any $t > 0$,

$$\Pr\Big(|Z - L| \geq t\Big) \leq \exp\Big(-\frac{t^2}{2\sum_{i=1}^{s} \mathbb{E}[X_i^2] + \frac{2}{3} M\, t}\Big).$$

Set $K = \|D\|_F^2 + k|D|$ so that $\gamma(\|D\|_F^2 + k|D|)\xi = \gamma K \xi$ and set

$$t := \varepsilon\Big(L + \gamma K \xi + \lambda\, \Phi_k(D)\Big).$$

Then,

$$\Pr\left(|Z - L| \geq \varepsilon\Big(L + \gamma K \xi + \lambda\, \Phi_k(D)\Big)\right) \leq \exp\left(-\frac{\varepsilon^2\big(L + \gamma K \xi + \lambda\, \Phi_k(D)\big)^2}{2\frac{(L+\gamma K \xi + \lambda\, R)^2}{s} + \frac{2}{3}\frac{L+\gamma K \xi + \lambda\, R}{s}\, \varepsilon\big(L + \gamma K \xi + \lambda\, \Phi_k(D)\big)}\right).$$

By choosing

$$s = \left\lceil \varepsilon^{-2}\Big(2 + \frac{2\varepsilon}{3}\Big)\right\rceil,$$

the exponent can be made sufficiently large so that the probability of failure is below $0.1$. That is, with probability at least $0.9$,

$$\left|\sum_{x \in D} \ell(x) - \sum_{x \in S} w(x)\, \ell(x)\right| \leq \varepsilon\Big(L + \gamma K \xi + \lambda\, \Phi_k(D)\Big).$$

$\square$

**Theorem 2.3** (Coreset Guarantee for Loss Approximation via Row Subset Selection). *Let $D \subseteq \mathbb{R}^m$ be a dataset of $n$ points with an embedding $E$, and suppose the loss function $\ell$ satisfies [Assumption 2.1](#) with constants $\gamma, \lambda > 0$. Let*

$$\Phi_k(D) = \min_{\substack{C \subseteq D \\ |C|=k}} \min_{A \in \mathbb{R}^{k \times m}} \|D - CA\|_F^2$$

*denote the optimal row subset selection cost using $k$ rows from $D$.*

*Then there exists a randomized algorithm that constructs a weighted subset $S \subseteq D$ of size $s = \mathcal{O}\left(\frac{1}{\varepsilon^2}\right)$ with weights $w(x)$ such that, with probability at least $0.9$,*

$$\left|\sum_{x \in D} \ell(x) - \sum_{x \in S} w(x)\, \ell(x)\right| \leq \tag{2}$$

$$\varepsilon \cdot \left(\sum_{x \in D} \ell(x) + \gamma\|D\|_F^2 + \gamma k|D| \max \ell + 2\lambda \cdot \Phi_k(D)\right).$$

*Equivalently, the weighted average loss over the selected rows $S$ approximates the true average loss over $D$ to within a $(1 \pm \varepsilon)$ factor, up to an additive term proportional to $\Phi_k(D)/n$.*

*Proof.* Let $L := \sum_{x \in D} \ell(x)$ denote the total loss over the dataset $D$. We define

$$\Phi_k(D) = \min_{\substack{C \subseteq D \\ |C|=k}} \min_{A \in \mathbb{R}^{k \times m}} \|D - CA\|_F^2$$

to be the optimal row subset selection cost using $k$ rows from $D$. For each $x \in D$, let $v(x) = \mathrm{Proj}(x, V)$ denote the projection of $x$ onto the subspace spanned by the selected rows $V$. Similarly, let $r(x) = x - \mathrm{Proj}(x, V)$ denote the

component of $x$ orthogonal to the subspace spanned by $V$, so that $x = v(x) + r(x)$. By the Lipschitz assumption on the loss function,

$$|\ell(x) - \ell(v(x))| \leq \lambda \cdot \|r(x)\|_2^2.$$

We decompose the component of $x$ within $V$ by $v(x) = \alpha_1 v_1 + \ldots + \alpha_t v_t$, so that

$$\left|\ell(v(x)) - (\alpha_1^2 \ell(v_1) + \ldots + \alpha_t^2 \ell(v_t))\right| \leq \gamma \left(|\alpha_1^2 - 1|\ell(v_1) + \ldots + |\alpha_t^2 - 1|\ell(v_t)\right).$$

By triangle inequality,

$$\ell(x) \leq (\alpha_1^2 \ell(v_1) + \ldots + \alpha_t^2 \ell(v_t)) + \gamma \left(|\alpha_1^2 - 1|\ell(v_1) + \ldots + |\alpha_t^2 - 1|\ell(v_t)\right) + \lambda\|r(x)\|_2^2$$
$$\leq (\gamma + 1)(\alpha_1^2 \ell(v_1) + \ldots + \alpha_t^2 \ell(v_t)) + \gamma t \cdot \max_t \ell(v_t) + \lambda\|r(x)\|_2^2.$$

Now, for a parameter $\xi \geq \max_t \ell(v_t)$, we define the sensitivity score for each $x \in D$ by

$$\sigma(x) := (\gamma + 1)(\alpha_1^2 \ell(v_1) + \ldots + \alpha_t^2 \ell(v_t)) + \gamma t \xi + \lambda\|r(x)\|_2^2,$$

and assign the sampling probability $p(x)$ for each $x \in D$ by normalizing the sensitivity scores, so that

$$p(x) = \frac{\sigma(x)}{\sum_{z \in D} \sigma(z)},$$

and thus $\sum_{x \in D} p(x) = 1$.

Consider the process of sampling $s$ independent samples with replacement from $D$ by the probability distribution induced by $p(x)$. Let $S = \{y_1, \ldots, y_s\}$ be denote the resulting multi-set and for each sample $y \in S$, define the resulting weight $w(y) = \frac{1}{s \cdot p(y)}$. Due to this weighting, the loss estimator is $Z = \sum_{y \in S} w(y)\ell(y)$ and thus by the linearity of expectation, the estimator is unbiased:

$$\mathbb{E}[Z] = \sum_{y \in S} \mathbb{E}[w(y)\ell(y)] = \sum_{y \in S} \sum_{x \in D} p(x) \cdot \frac{\ell(x)}{s \cdot p(x)} = \sum_{y \in S} \frac{L}{s} = L.$$

We next analyze the variance of the estimator. We have

$$\mathbb{E}[y^2] = \sum_{x \in D} p(x) \left(\frac{\ell(x)}{s \cdot p(x)}\right)^2 = \frac{1}{s^2} \sum_{x \in D} \frac{\ell(x)^2}{p(x)}.$$

Writing $T = \sum_{z \in D} \sigma(z)$ so that $p(x) = \frac{\sigma(x)}{T}$, then we have

$$\mathbb{E}[y^2] = \frac{T}{s^2} \sum_{x \in D} \frac{\ell(x)^2}{(\gamma + 1)(\alpha_1^2 \ell(v_1) + \ldots + \alpha_t^2 \ell(v_t)) + \gamma t \xi + \lambda\|r(x)\|_2^2}.$$

By assumption, the loss function satisfies $\ell(x) \leq (\gamma + 1)(\alpha_1^2 \ell(v_1) + \ldots + \alpha_t^2 \ell(v_t)) + \gamma t \xi) + \lambda\|r(x)\|_2^2$ and thus

$$\frac{\ell(x)^2}{(\gamma + 1)(\alpha_1^2 \ell(v_1) + \ldots + \alpha_t^2 \ell(v_t)) + \gamma t \xi + \lambda\|r(x)\|_2^2} \leq \ell(x).$$

Hence,

$$\mathbb{E}[y^2] \leq \frac{T}{s^2} \sum_{x \in D} \ell(x) = \frac{LT}{s^2}.$$

Then over the $s$ samples, it follows that

$$\sum_{i=1}^{s} \mathbb{E}[y_i^2] \leq \frac{LT}{s}.$$

Since we have $T \leq L + \gamma(\|D\|_F^2 + t \cdot |D|)\xi + \lambda R$, then

$$\sum_{i=1}^{s} \mathbb{E}\left[y_i^2\right] \leq \frac{\left(L + \gamma(\|D\|_F^2 + t \cdot |D|)\xi + \lambda R\right)^2}{s}.$$

Finally, before we can apply standard concentration inequalities, we also need to upper bound the maximum of each weighted contribution. To that end, observe that for any point $x \in D$, its weighted contribution is

$$\ell(x)w(x) = \frac{\ell(x)}{s}\frac{1}{p(x)} = \frac{\ell(x)}{s}\frac{T}{\sigma(x)}.$$

Because $\ell(x) \leq \sigma(x)$, we have

$$\ell(x)w(x) \leq \frac{T}{s} \leq \frac{L + \gamma(\|D\|_F^2 + t \cdot |D|)\xi + \lambda R}{s}.$$

Thus, it follows that $|y_i| \leq M$ for every sample, for

$$M := \frac{L + \gamma(\|D\|_F^2 + t \cdot |D|)\xi + \lambda R}{s}.$$

By applying Bernstein's inequality to the estimator $Z = \sum_{i=1}^{s} X_i = \sum_{x \in S} w(x) \cdot \ell(x)$, we have that for any $r > 0$,

$$\mathbf{Pr}\left[(|Z - L| \geq r)\right] \leq \exp\left(-\frac{r^2}{2\sum_{i=1}^{s} \mathbb{E}\left[y_i^2\right] + \frac{2}{3}Mr}\right).$$

Let $K = \|D\|_F^2 + t \cdot |D|$ so that $\gamma(\|D\|_F^2 + t \cdot |D|)\xi = \gamma K\xi$ and set

$$t := \varepsilon \cdot (L + \gamma K\xi + \lambda\,\Phi_k(D)).$$

Then,

$$\mathbf{Pr}\left[|Z - L| \geq \varepsilon\Big(L + \gamma K\xi + \lambda \cdot \Phi_k(D)\Big)\right] \leq \exp\left(-\frac{\varepsilon^2\left(L + \gamma K\xi + \lambda \cdot \Phi_k(D)\right)^2}{2\frac{(L+\gamma K\xi + \lambda \cdot R)^2}{s} + \frac{2}{3}\frac{L+\gamma K\xi + \lambda \cdot R}{s} \cdot \varepsilon\left(L + \gamma K\xi + \lambda \cdot \Phi_k(D)\right)}\right).$$

By choosing

$$s = \left\lceil \varepsilon^{-2}\left(2 + \frac{2\varepsilon}{3}\right)\right\rceil,$$

the exponent can be made sufficiently large so that the probability of failure is below $0.1$. That is, with probability at least $0.9$,

$$\left|\sum_{x \in D} \ell(x) - \sum_{x \in S} w(x) \cdot \ell(x)\right| \leq \varepsilon \cdot (L + \gamma K\xi + \lambda \cdot \Phi_k(D)).$$

Finally, the claim follows by taking a bicriteria algorithm so that $t = \mathcal{O}(k)$ and rescaling $\varepsilon$ as necessary. $\qquad\square$

## D. Empirical Validation of Assumption 2.1

We verify Assumption 2.1 directly on the Credit Card logistic regression task by evaluating both sides of each inequality on every sample and plotting the left-hand side against the right-hand side. The theoretical bound corresponds to the diagonal LHS = RHS: points below the diagonal satisfy the inequality, and the vertical gap to the diagonal measures the slack. Figure 5 shows the result for $\lambda = 0.3$ and $\gamma = 5$. In both panels, all empirical points lie well below the diagonal, confirming that the assumption holds across the dataset with substantial slack at these constants.

# E. Conclusion

In this work, we introduced a novel data selection framework based on low-rank approximation, diverging from traditional clustering methods. We proposed a sensitivity sampling algorithm that constructs a small, weighted coreset to approximate the loss of the full dataset. Our main theoretical result, Theorem 2.2, provides a rigorous guarantee for this approach, with an error bound directly tied to the dataset's alignment with a low-rank structure.

Our empirical evaluations confirmed the practical benefits of this method. Across both a standard tabular dataset and challenging Llama3-8B fine-tuning on three tasks, our low-rank approach outperformed uniform sampling and clustering-based techniques in both approximation quality and downstream model performance. Our work provides a scalable, theoretically-grounded, and effective solution for data-efficient training, offering a robust alternative by leveraging the low-rank structure of data to identify the most informative samples.

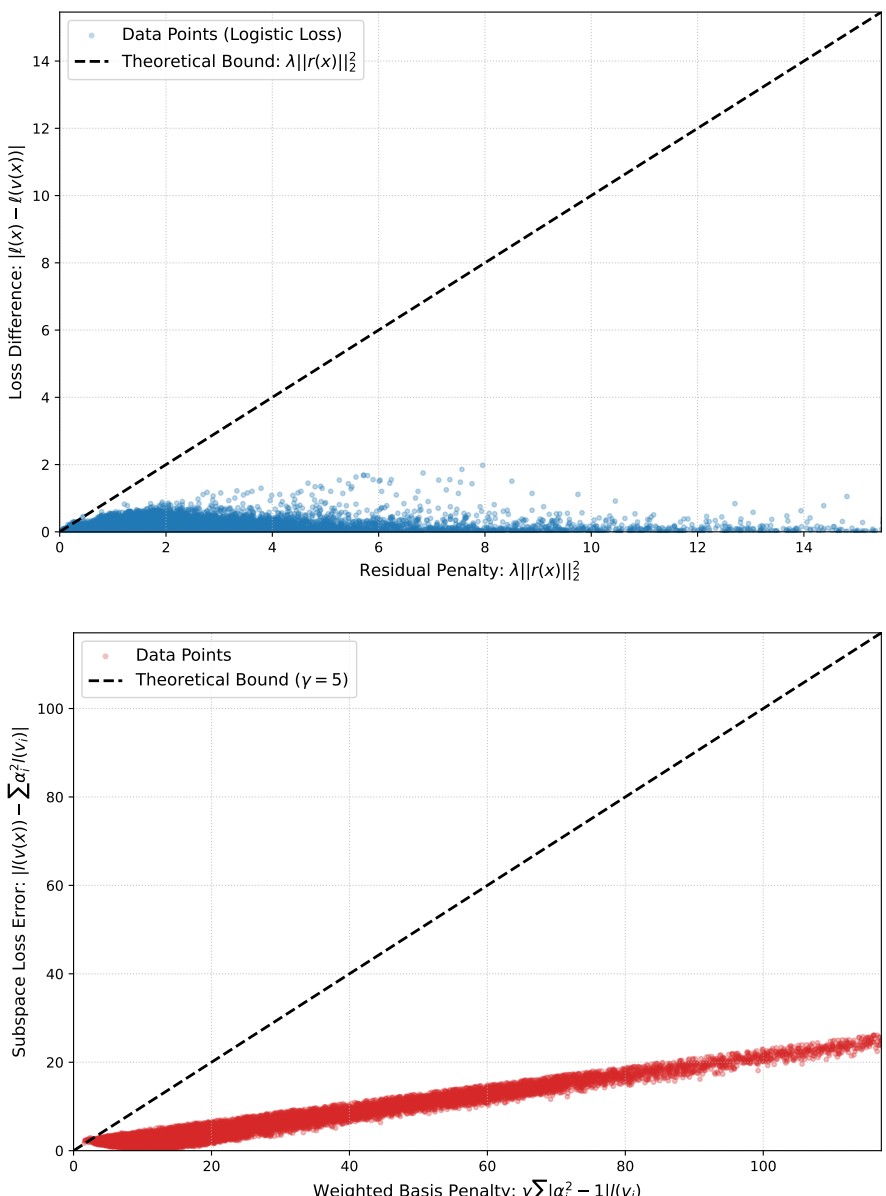

*Figure 5.* Empirical validation of Assumption 2.1 on the Credit Card task. **Top:** $|\ell(x) - \ell(v(x))|$ vs. $\lambda\|r(x)\|_2^2$ with $\lambda = 0.3$. **Bottom:** $|\ell(v(x)) - \sum_i \alpha_i^2 \ell(v_i)|$ vs. $\gamma \sum_i |\alpha_i^2 - 1|\ell(v_i)$ with $\gamma = 5$. Empirical points lie below the theoretical bound (dashed line) in both cases.

