# OpenReview forum: "Active Learning with Low-Rank Structure for Data Selection"
_ICML.cc/2026/Conference — ICML 2026 regular_

### Official Review · Reviewer_zfPU · 2026-03-11

**Soundness:** 2
**Presentation:** 2
**Significance:** 2
**Originality:** 3
**Overall Recommendation:** 4
**Confidence:** 3

**Summary:**

This paper proposes a data selection method based on the low-rank structure of the loss landscape. The goal is to construct a weighted subset (coreset) that can approximate the information contained in the full dataset, thereby reducing computational cost in large-scale training and related tasks. The key idea is to exploit a low-rank decomposition of the loss structure and use this structure to guide the sampling of representative data points. In particular, the method assigns importance scores to samples based on their projections onto a low-rank subspace as well as their residual components, and then constructs a weighted subset that approximates the empirical loss over the entire dataset. The paper also provides theoretical guarantees for the approximation quality of the constructed subset under certain assumptions. Overall, the work aims to provide a principled framework for loss-aware data selection with theoretical justification and potential applications to large-scale training and active learning settings.

**Compliance With Llm Reviewing Policy:**

Affirmed.

**Final Justification:**

This paper proposes a principled framework for constructing weighted data subsets by exploiting the low-rank structure of loss variations. The idea is novel, connecting coreset construction with loss approximation, and is supported by theoretical guarantees.

My initial concerns focused on the practical validity of the low-rank assumption, computational overhead, model-dependence of selected subsets, and notational clarity.

The authors' rebuttal satisfactorily addresses these points:

They provide empirical validation (Sec. 4.2.2) showing the assumption holds well in their fine-tuning context.

A new cross-model experiment demonstrates that subsets selected by one LLM effectively train a different architecture, alleviating model-dependence concerns.

They clarify that computational costs are manageable (SVD on GPU is negligible; loss evaluation is a small fraction of training).

They commit to standardizing notation throughout.

While the theory relies on assumptions that may not hold in all regimes, the empirical evidence in the relevant setting is convincing. The cross-model results particularly strengthen the case for practical utility.

I therefore recommend Weak Accept (4). The paper makes a clear contribution, and the promised revisions will further improve the manuscript.

**Key Questions For Authors:**

1.Under what conditions does Assumption 2.1 hold in practice? When the assumption is satisfied, how should the parameters λ and γ in Algorithm 2.1 be chosen?
2.The construction of the weighted subset in this paper depends on a specific model (e.g., model A), since it relies on the loss under the current model l(x,y;A). In Section 3.1, the authors claim that the coreset constructed using the current model loss can approximate the full dataset loss during subsequent training. However, during training the model parameters will continue to evolve, which may change the loss landscape. As a result, samples that were initially important may become less important later, while some initially unimportant samples may become more important. In such cases, the originally selected coreset may become less representative of the full dataset. How does the proposed method address this potential issue?
3.The experimental setup in Section 4.1 may not be entirely fair. The authors first train a model on the training set and compute the loss to obtain scores, and then construct a weighted subset based on these scores. In other words, the data selection process is explicitly designed based on the model’s loss information, whereas some baselines may not have access to this information. The authors should present a more complete workflow, for example by performing data selection during intermediate stages of training and then continuing training on the selected subset. It would also be helpful to report the computational overhead of the selection stage for different methods.
4.The paper also mentions active learning as a potential application. However, the proposed method requires computing loss values l(x,y;A), which depend on the ground-truth labels. In a typical active learning setting, the candidate data pool is unlabeled, and the loss cannot be directly computed. It would be helpful if the authors clarify how the proposed framework applies to the active learning scenario, and whether the theoretical guarantees still hold in that case.

**Limitations:**

See weaknesses and questions.

**Strengths And Weaknesses:**

*Strengths:*

1.The paper proposes a data selection strategy that differs from many existing approaches. The idea of exploiting the low-rank structure of the loss (or its residual components) to guide data selection is reasonable and conceptually interesting. The construction of a weighted subset to approximate the full dataset is also an appealing idea.

2.The paper provides theoretical guarantees for the proposed method rather than relying purely on heuristics. The overall structure of the paper is complete, and the algorithm is supported by formal analysis.

3.The proposed framework connects ideas from data selection, coreset construction, and loss approximation, which may provide useful insights for large-scale training and related problems where training on the entire dataset is computationally expensive.

*Weakness:*

1.The theoretical results rely heavily on Assumption 2.1, and the paper only briefly mentions situations where the assumption might hold (e.g., Lipschitz continuity). However, in many practical scenarios, especially in modern deep learning, the loss landscape can be highly nonlinear and complex, and the assumption may not hold. The paper does not further discuss the practical validity of this assumption, nor does it empirically verify whether the assumption approximately holds in real datasets. As a result, the theoretical guarantees may mainly apply to idealized settings.

2.The experimental evaluation is not sufficiently comprehensive. In Algorithm 2.1, the construction of the weighted subset may involve operations such as computing leverage scores, performing SVD, and evaluating losses. When the dataset D is large, these steps may introduce nontrivial computational overhead. However, the experimental section provides little discussion of these costs. The authors should report the computational time and complexity of the sampling stage for the proposed method and the baselines, in order to provide a more complete evaluation of the practical efficiency of the method.

3.The construction of the weighted subset is model-dependent, since it relies on the loss values under a specific model l(x,y;A). As a result, the selected subset is tailored to the current model used during selection. It is unclear whether such a subset would remain effective if the model architecture or training configuration changes. This raises questions about the generality and robustness of the selected coreset across different models.

4.The use of mathematical notation and formulas in the paper is somewhat inconsistent and occasionally confusing. For example, the symbol D is used both to denote a dataset (as a set) and to represent a data matrix. In Theorem 2.2, the dimension of D appears to be n*m, while in Theorem 2.3 it appears to be m*n. These inconsistencies affect the readability of the paper. The authors should standardize the use of mathematical notation and clarify the definitions of symbols.

---

> ### Author Rebuttal · Authors · 2026-03-31
>
> > 1...Assumption 2.1...may mainly apply to idealized settings.
>
> We empirically validate Assumption 2.1 in Section 4.2.2. On GSM8K with BERT embeddings, the low-rank approximation closely matches the true loss and significantly outperforms clustering, indicating that dominant spectral directions capture most loss variation in practice, meaning the assumption holds sufficiently well in practical settings to yield high-quality coresets.
>
> > 2... Algorithm 2.1...may involve operations...introduce nontrivial computational overhead...should report the computational time...for the proposed method and the baselines
>
> In practice, the runtime for computing (low-rank) SVD in the embedding space and the leverage scores, which we perform on GPU, is negligible compared to performing training. Moreover, computing loss values is more expensive, as the reviewer suggests. However, as discussed in page 7 of the paper, it only takes around 6.67% of a single-epoch training, since it only needs forward passes on k=20% of the data.
>
> > 3....the weighted subset is model-dependent...questions about the generality and robustness...across different models.
>
> We address the concern on model-dependence via a cross-model transfer experiment (see response to Q3). We selected ViGGO subsets using Llama-3-8B-Instruct and used them to train a different architecture, Qwen-2.5-3B-Instruct; our method still significantly outperforms baselines. This indicates the selected subsets capture intrinsic, model-agnostic dataset difficulty and information structure, rather than overfitting to the selection model.
>
> > 4....mathematical notation...inconsistent
>
> Thanks, we have made another pass through the manuscript, carefully standardizing the notation, clearly separating datasets (as sets) from their matrix representations and eliminating the previous overloading of $D$. We also fixed a consistent convention for matrix dimensions and orientation, stated at first use and followed uniformly, resolving the discrepancies between Theorems 2.2 and 2.3.
>
> > 1...Assumption 2.1 hold in practice?...how should...λ and γ...be chosen?
>
> Assumption 2.1 holds when embeddings are sufficiently informative so that loss variation aligns with their principal components—typical for dense pretrained representations (e.g., BERT, GTR).
>
> For parameters, as discussed in Section 4.2.1, $\gamma$ controls the low-rank approximation error; in our LLM fine-tuning setup we set $\gamma=0$ and compute coefficients via KRR. The parameter $\lambda$ weights the residual term and is tuned; Figure 3 shows a stable range (e.g., $[0.01,1]$), while very large values (e.g., $>100$) hurt performance by overemphasizing outliers. We will include clearer guidelines in the appendix.
>
> > 2....during training the model parameters will continue to evolve, which may change the loss landscape...potential issue?
>
> Although the loss landscape evolves during training, we focus on fine-tuning pre-trained foundation models, where parameter updates are relatively small, and so, the relative importance and low-rank structure of data points remain largely stable over a single epoch. This is supported by our new cross-model transfer experiment (see response to Q3), where a subset selected from Llama’s initial loss landscape effectively trains a Qwen model from scratch. Moreover, Figure 4 shows strong performance when using gradient norms, which capture training dynamics. In settings with larger shifts (e.g., training from scratch), our framework naturally supports periodic resampling via updated low-rank sensitivity scores.
>
> > 3....data selection process is...based on the model’s loss information, whereas some baselines may not have access to this information....
>
> We took the ViGGO subsets selected by the Llama-3-8B-Instruct model from the paper, and trained a Qwen-2.5-3B-Instruct on them. Below are the final accuracy numbers for 25% and 12.5% compression rates for Clustering-based SS and Low-rank SS (ours). We also included other baselines, and overall the results indicate the selected subset transfers nicely from Llama to Qwen.
>
> | Method | 25% Compression | 12.5% Compression |
> | :--- | :--- | :--- |
> | Uniform | 68.35 | 45.38 |
> | K-Center | 59.24 | 43.14 |
> | Clustering-based SS (selection with Qwen) | 71.57 | 48.46 |
> | Lowrank SS (ours, selection with Qwen) | 75.91 | 51.54 |
> | Clustering-based SS (selection with Llama) | 72.27 | 47.06 |
> | Lowrank SS (ours, selection with Llama) | 76.05 | 55.74 |
>
> > 4....In a typical active learning setting...the loss cannot be directly computed...
>
> In a strict active learning setting, our framework applies in two ways: (1) query only the $k$ landmarks, since $\sigma(x)$ depends only on $\ell(v_1),\ldots,\ell(v_k)$, enabling computation of $p(x)$ without labels for the full pool; (2) replace the supervised loss with a proxy (e.g., uncertainty or gradient norm), as validated in Section 4.2.2 (Figure 4). In both cases, Theorems 2.2–2.3 guarantee preservation of the chosen objective.

---

> > ### Author Rebuttal · Reviewer_zfPU · 2026-04-03
> >
> > Thanks for addressing my raised concerns. I hope the authors could add the explanations and experiments on theoritical assumptions, model transfer and clearer guidelines in the manuscript. I will consider upgrading my score.

---

> > > ### Author Response · Authors · 2026-04-07
> > >
> > > Thank you for your follow-up. We’re glad our rebuttal addressed your concerns about the theoretical assumptions, cross-model transfer, and practical guidelines, and we’ve incorporated these updates into the revised manuscript. If you feel that these points have been fully resolved, we would be grateful if you considered updating your score accordingly. We appreciate your positive comments and detailed feedback.

---

### Official Review · Reviewer_q2Ww · 2026-03-13

**Soundness:** 3
**Presentation:** 4
**Significance:** 3
**Originality:** 3
**Overall Recommendation:** 5
**Confidence:** 3

**Summary:**

The authors study an algorithm for active learning and core set selection based on the principle of low-rank approximation. Their framework relies on one key assumption: that the dominant directions of variance capture most of the loss, while the orthogonal directions contribute only abounded amount. The proposed strategy first involves constructing a sketch of the dataset. The sketch is used to determine the sampling distribution on the dataset which is characterized by a sensitivity score for each point, reflecting how much samples contribute to the total loss relative to their projection and residuals. The authors provide a few results to justify their method. In particular, the sampling complexities for when the core set is a subset of the data or not. A variety of experiments are also provided including 1) an evaluation of the core set and test error on the credit card dataset compared to randomized and cluster-based coreset selection 2) LLM fine-tuning experiments on three challenging datasets. These experiments indicate competitive results compared to clustering-based methods.

**Compliance With Llm Reviewing Policy:**

Affirmed.

**Final Justification:**

authors addressed my concerns in their rebuttal.

**Key Questions For Authors:**

see weaknesses

**Limitations:**

yes

**Strengths And Weaknesses:**

**strength** This is a comprehensive and well-written manuscript. In most places, it is easy to read, and overall cleverly addresses an important problem in machine learning (active learning) via an existing framework (low-rank coreset selection / row subset selection) and provides some theoretical justification for its application.

**weakness** I feel that the experiments are the weakest part of the paper - for example, the authors do not empirically justify the central assumption, which is not obviously satisfied by real-world data. Perhaps the authors can demonstrate that neural net embeddings or generic datasets satisfy it? Also, the authors claim a generic sketch can be used to bootstrap the coreset construction, but it is not obvious to me (maybe I missed something) how different algorithms influence the performance of the algorithm. Could the authors address these concerns?

---

> ### Author Rebuttal · Authors · 2026-03-31
>
> > weakness I feel that the experiments are the weakest part of the paper - for example, the authors do not empirically justify the central assumption, which is not obviously satisfied by real-world data. Perhaps the authors can demonstrate that neural net embeddings or generic datasets satisfy it?
>
> We thank the reviewer for raising this important point regarding the empirical validation of Assumption 2.1. We would like to point out that we do empirically justify our central assumption in two contexts:
> - The LLM Setting: In our LLM fine-tuning experiments, computing the exact loss $\ell(v(y))$ is intractable. The projection $v(y)$ exists in the continuous BERT embedding space and cannot be decoded into a discrete text sample to evaluate the LLM loss. As noted in the paper, "Assumption 2.1 should be interpreted as a regularity and approximation condition rather than an exact structural requirement on the loss function." The strong end-to-end performance of our derived algorithm across multiple NLP tasks validates the practical utility of this assumption.
> - Credit Card Experiments (logistic loss): We can directly verify the assumption in this context. Using the exact logistic regression setup for the Credit Card dataset (Section 4.1), we computed both inequalities of Assumption 2.1. We plotted the Left-Hand Side (LHS) on the y-axis against the Right-Hand Side (RHS) on the x-axis for samples across the dataset. As shown in the two figures linked at https://imgur.com/a/hmghfLN (using $\lambda=0.3$ and $\gamma=5.0$), the bounding behavior cleanly holds, providing direct empirical support for our theoretical framework.
>
> Taken together, we believe this provides good empirical evidence that real-world neural embeddings inherently exhibit the algebraic structure required to satisfy Assumption 2.1. We will explicitly tie Figure 2 to Assumption 2.1 in the text to make this connection clearer.
>
> > Also, the authors claim a generic sketch can be used to bootstrap the coreset construction, but it is not obvious to me (maybe I missed something) how different algorithms influence the performance of the algorithm. Could the authors address these concerns?
>
> The theoretical guarantees of our framework (Theorems 2.2 and 2.3) depend directly on $\Phi_k(D)$, which is the approximation cost of the sketch itself. As long as the sketch algorithm provides a bi-criteria approximation (as noted in Theorem 3.1), the theoretical guarantees hold. In practice, the choice of sketch presents a trade-off between computational speed and the tightness of the bound (via the $\Phi_k(D)$ term). In our experiments, we used exact Truncated SVD for tabular data and highly scalable leverage score sampling for LLMs. We will clarify this connection between the sketch algorithm, the $\Phi_k(D)$ cost, and empirical performance in the revised text.
>
> Empirically, we repeated the ViGGO 12.5% experiments on Qwen-2.5-3B-Instruct, but instead of leverage score sampling, we tried (1) selectiong the basis samples uniformly at random, and (2) setting the basis as the result of a running the k-medians algorithm. The accuracy numbers are reported below. The results show that leverage score sampling has an edge compared to uniform selection, while both outperform k-medians by a notable margin.
>
> | Method | Uniform Basis| K-Medians | Leverage Score |
> | :--- | :--- | :--- | :--- |
> | ViGGO 12.5% Accuracy| 50.70 | 45.80 | 51.54 |

---

> > ### Author Rebuttal · Reviewer_q2Ww · 2026-04-02
> >
> > Thanks for carefully responding to my review - particularly my questions regarding the assumptions. I have no additional questions at this time. I will update my score.

---

> > > ### Author Response · Authors · 2026-04-07
> > >
> > > Thank you for the additional follow-up. We’re glad the rebuttal helped clarify your questions around the empirical support for our assumptions and the role of the sketching step. We really appreciate your positive feedback and your careful reading of the paper.

---

### Official Review · Reviewer_qdzg · 2026-03-14

**Soundness:** 2
**Presentation:** 3
**Significance:** 3
**Originality:** 2
**Overall Recommendation:** 4
**Confidence:** 3

**Summary:**

In this paper, a new data selection algorithm, called low-rank SS, is proposed. The proposed algorithm leverages low rank approximation and sensitivity sampling to create the representative subset of data. It also provides the proofs for the proposed theorems. The experimental results show that the proposed algorithm can achieve comparable or better performance than existing methods.

**Compliance With Llm Reviewing Policy:**

Affirmed.

**Final Justification:**

All my remaining questions have been fairly resolved, so I will maintain my original recommendation.

**Key Questions For Authors:**

1. The datasets used in the experiments are relatively small, which makes it difficult to assess the scalability and practical impact of the proposed method. Additional experiments on larger-scale datasets would help better understand its effectiveness in practical training settings.

2. The experiments only use BERT embeddings, making it unclear how robust the method is to the choice of embedding model. It would be helpful to evaluate the approach with other embeddings to show its generalizability.

3. In Section 4.2, the LLM fine-tuning experiments are conducted only with LLaMA3-8B model, even though these are core experiments to show the effectiveness of the proposed algorithm. Therefore, evaluating the proposed algorithm with other LLM would strengthen the generality of the results.

**Limitations:**

yes

**Strengths And Weaknesses:**

**Strength**
1. This paper is well written and easy to follow.
2. This paper provides mathematical proofs for the proposed theorems.
3. The experimental results demonstrate the effectiveness of the proposed algorithm.

**Weakness**
1. The experimental results may not be sufficient to fully demonstrate the effectiveness and generalizability of the proposed algorithm.

---

> ### Author Rebuttal · Authors · 2026-03-31
>
> > The experimental results may not be sufficient to fully demonstrate the effectiveness and generalizability of the proposed algorithm.
>
> > The datasets used in the experiments are relatively small, which makes it difficult to assess the scalability and practical impact of the proposed method. Additional experiments on larger-scale datasets would help better understand its effectiveness in practical training settings.
>
> We thank the reviewer for their comments. We would like to clarify that the datasets used in our LLM fine-tuning experiments (Section 4.2) are standard benchmarks for instruction tuning and represent realistic scales for this domain. Specifically, we fine-tune a Llama3-8B model on GSM8k (7.47k samples), ViGGO (5.1k samples), and SQL (30k samples). This experimental setup matches recent leading works on large language model fine-tuning and data selection (e.g., HALO [Ashkboos et al., 2025]). These datasets and models are sufficiently complex to demonstrate the practical impact and effectiveness of our method on modern architectures.
>
> Furthermore, our approach is inherently scalable: computing the low-rank approximation (e.g., leverage scores and truncated SVD) in the embedding space is computationally lightweight and easily scales to much larger datasets without bottlenecking the training pipeline. We will emphasize the scalability and the representativeness of these dataset sizes in the revised text.
>
> > The experiments only use BERT embeddings, making it unclear how robust the method is to the choice of embedding model. It would be helpful to evaluate the approach with other embeddings to show its generalizability.
>
> We would like to point out that Figure 4 already includes experiments with GTR embeddings [1] as well as BERT. Our results indicate that our findings are consistent on both embedding models.
> [1] https://huggingface.co/sentence-transformers/gtr-t5-base
>
> > In Section 4.2, the LLM fine-tuning experiments are conducted only with LLaMA3-8B model, even though these are core experiments to show the effectiveness of the proposed algorithm. Therefore, evaluating the proposed algorithm with other LLM would strengthen the generality of the results.
>
> We thank the reviewer for their comment. To address their concern, we repeat the ViGGO experiments with the Qwen-2.5-3B-Instruct model, and compare our method with the baselines in this setting. The same hyperparameters as the paper are used, except that the learning rate was increased to 1e-4.
> Below please find the accuracy numbers for 25% and 12.5% compression rates. These results clearly indicate the improvements achieved by our method.
>
> | Method | 25% Compression | 12.5% Compression |
> | :--- | :--- | :--- |
> | Uniform | 68.35 | 45.38 |
> | K-Center | 59.24 | 43.14 |
> | Graph Cut | 62.89 | 38.94 |
> | Clustering-based SS | 71.57 | 48.46 |
> | Lowrank SS (ours) | 75.91 | 51.54 |
>
> We appreciate the reviewer’s comment and will include the new results in the next revision of the paper.

---

> > ### Author Rebuttal · Reviewer_qdzg · 2026-04-04
> >
> > I appreciate the authors’ response. I will keep my positive score.

---

> > > ### Author Response · Authors · 2026-04-07
> > >
> > > Thank you for your prompt follow-up. We are glad that our rebuttal has fully resolved your concerns regarding dataset scale, embedding robustness, and generalization across LLMs. We appreciate your positive assessment.

---

### Decision · Program_Chairs · 2026-04-30

**Decision:**

Accept (regular)

**Comment:**

All reviewers agree on the technical and theoretical contributions of this paper. The authors have successfully addressed all the concerns of the reviewers. Hence, I recommend accepting this paper.